# Patient perceptions of the impact of inducible laryngeal obstruction on quality of life

**Katherine M. McConville** [1]*, **Susan L. Thibeault**[2]

**1** Department of Speech-Language Pathology, Michigan Medicine, The University of Michigan, Ann Arbor, Michigan, United States of America, **2** Division of Otolaryngology—Head and Neck Surgery, School of Medicine and Public Health, University of Wisconsin-Madison, Madison, Wisconsin, United States of America

\* katmccon@med.umich.edu

## Abstract

### Background

Inducible laryngeal obstruction (ILO) accounts for or contributes to dyspnea in a noteworthy proportion of treatment seeking populations including those misdiagnosed with asthma. Despite increasing awareness of the disorder, literature exploring patient experience is limited. The aim of this work is to report patient perspectives on ILO and the way in which it impacts quality of life.

### Methods

This qualitative study utilized methods detailed in the literature on grounded theory and phenomenological research to analyze interviews collected from participants diagnosed with ILO. Interviews were conducted, audio recorded, and transcribed. Transcriptions underwent content-analysis using Burnard's 14 step method [15], which included review of content codes across multiple raters until consensus regarding analyses was reached.

### Results

Twenty-six participants were included in the study. Most participants were female (92%). Ages ranged from 18–72 with a mean age of 45 for female participants and 37 for male participants. Without specific prompting to do so, all participants offered descriptions of the specific symptoms they experienced and the triggers for their symptoms. In the content analysis process, "descriptions of symptoms and triggers" was thus labeled a theme that was present in all interviews. Seven additional themes were shared consistently and judged to encapsulate the interview material. These themes were: 2) *diagnosis and treatment*, 3) *emotional impact of ILO*, 4) *perception of health and prognosis*, 5) *ameliorating factors*, 6) *influence of ILO on lifestyle*, 7) *the physical impact of ILO*, and 8), *social consequences of ILO*. In addition, 54 subthemes were identified.

### Conclusions

Patients appear to place particular emphasis on the emotional and psychosocial consequences of ILO as well as factors that ameliorate the condition. As such, future efforts to

**Data Availability Statement:** Requests for data should be directed to the UW-Madison IRB (IRBdirector@hsirb.wisc.edu). All data cannot be shared publicly for ethical and legal reasons. The data are comprised of transcribed interviews that

may contain sensitive, identifiable, patient information. Data requests can be directed to UW Madison IRB (contact via IRBdirector@hsirb.wisc.edu) for researchers who meet the criteria for access to confidential data.

**Funding:** The authors have declared that no financial disclosures exist.

**Competing interests:** The authors have declared that no competing interests exist.

treat ILO and to collect outcomes measures should account for these aspects of the patient experience.

## Introduction

Inducible laryngeal obstruction (ILO) is a laryngeal disorder characterized by the unwanted adduction of the vocal folds, temporarily restricting the laryngeal airway, resulting in difficulty breathing and related symptoms of discomfort [1]. Considered a diagnosis of elimination, patient report of symptoms in the absence of other etiologies is common in supporting a diagnosis of ILO. Other clinical findings like truncated inspiratory flow loops during pulmonary function tests have been suggested by some to indicate the presence of ILO [2,3] but an indirect laryngeal exam when symptoms are present is the current gold standard in diagnosis [4]. Dyspnea, the chief symptom of ILO, has long been inversely associated with quality of life (QOL) in patients with COPD, and congestive heart failure [5,6]. While dyspnea can result in patient morbidity and distress irrespective of etiology, studies relating spirometry and laryngeal exams to the severity of ILO have not been pursued, likely because these measures are not thought to be consistently sensitive to the episodic presentation of the disorder [7], and they do not delineate an objective level of breathing impairment in the same way that pulmonary function may provide benchmarks for staging chronic conditions such as COPD. More recently, work has focused on the cost associated with care of dyspnea prior to and following diagnosis of ILO [8]. Additionally, questionnaires focused on upper airway disorders [9] and/or ILO-specific items [10] have been developed to quantify patient perception of impairment which may prove beneficial in future outcomes-related research on this disorder. Yet, the specific impact of ILO on quality of life has not been closely examined and reported via direct patient report. It is important to systematically study and document patient perspectives regarding personal experiences, as the collective input from patients with this diagnosis could improve understanding and treatment of the disorder and help to refine the measurement of treatment outcomes.

The purpose of the present investigation was to document patient perspectives on ILO, providing a qualitative analysis detailing experiences that appear to be the most consistent and meaningful to patients with this disorder as it impacts their quality of life. Assessing patients' experience in this manner is important in reducing the bias that may occur when clinicians identify, and target treatment outcomes based mostly on their professional experience. Characterizing the salient experiences of patients with ILO will thus improve our ability to refine assessment and treatment in a way that may better support the specific needs of patients.

## Materials and methods

Individuals diagnosed with ILO in an outpatient clinic were recruited to participate in this qualitative study. Utilizing qualitative methods detailed in the literature on grounded theory and phenomenological research [15], participants underwent semi-structured interviews about their experiences; responses were transcribed and analyzed. Participants were recruited from a large tertiary care academic outpatient clinic from a variety of different specialties including primary care, pulmonology, and asthma/allergy. Assessment of laryngeal appearance and function was completed using videostroboscopy as standard of care at all evaluations. Given that ILO can be described as a disorder of laryngeal behavior or function, patients were

routinely evaluated and treated in our clinic by Speech-Language Pathologists (SLP). Occasionally, they were scheduled in multi-disciplinary visits with Otolaryngology (ENT).

### Participant recruitment and selection

Participants were recruited during clinic visits where they were approached by their treating clinician about eligibility. Additionally, recruitment flyers were posted in the waiting areas of the clinic prompting some participants to enquire about their eligibility with their treating clinicians. Eligible participants were at least 18 years old and had been diagnosed with ILO based on symptomatic laryngeal exam, clinical symptoms *and* history of spirometry with truncated inspiratory loop, or reported symptoms so consistent with ILO that assessing clinician provided a rating of confidence in the diagnosis equal to or exceeding 8/10. Those under the age of 18, or those without endoscopic or strong clinical evidence of ILO were excluded from the study. As ILO is considered a diagnosis of exclusion, meaning it is often reached by a process of elimination, and because it is known to co-occur frequently with a number of conditions [11], [12] those with comorbid asthma, allergy and COPD were not excluded from the study. It has been documented that many patients who either clearly have, or likely have ILO, still retain other diagnoses such as asthma while ILO is being diagnosed [13]. Participants presented with dyspnea as their *primary* complaint but were not excluded if they presented with secondary comorbid cough or voice quality concerns as these symptoms have been described as belonging to a spectrum of laryngeal sensitivity issues associated with ILO [14]. Participants had completed assessment including laryngeal visualization and varying degrees of treatment (from treatment not yet initiated to treatment completed).

### Semi-structured interviews

Interviews were conducted by speech language-pathologists who specialized in managing ILO. Interview questions were pre-scripted and could be modified/expanded when necessary to promote clarity of responses. Interview questions are listed in **Table 1**. All interviews were recorded in a quiet, private room. Full transcripts were typed by research assistants and provided to the initial rater, KM.

### Ethical considerations

This study was approved by the University of Wisconsin Minimal Risk Institutional Review Board (UW MR-IRB). Participants enrolled in the study provided written informed consent to participate in the recorded interviews and have their responses published. To minimize bias, participants were not interviewed by their treating clinicians and care was taken to phrase all questions in a neutral fashion.

### Data analysis

Burnard's 14 step method for transcript analysis [15] was employed as the basis for content analysis. This method involved 5 stages during which KM read the transcripts multiple times and then devised categories for all meaningful statements from the interviews conducted. Next, two additional raters with six to ten years of professional experience in research and/or clinical management of ILO were asked to categorize meaningful statements extracted from the interviews. A total of 8 overarching themes were agreed upon to address all content derived across participants. Categories were identified and the frequency with which these emerged in interviews is summarized in **Table 2**. A random sample of statements was pulled in increments of 10–20% of the total interview content until good rater consensus was reached. Agreement

**Table 1. Interview questions to assess patient experiences related to paradoxical vocal fold motion disorder.**

| Categories | Interview Questions | Follow up questions |
|---|---|---|
| General | | |
| | What do you experience?<br>How does this make you feel emotionally?<br>How does this make you feel physically? | Can you tell me more?<br>How do you react to this/what do you usually do? |
| Specific Impact | | |
| | Is any part of your lifestyle different as a result of your breathing?<br>Has this experience resulted in anything different for you at home or at work?<br>Do you experience a greater impact from you breathing at certain time of the day or in certain situations or environments? | How does this make you feel?<br>Can you tell me more?<br>What does that mean to you? |
| Influencing Factors | | |
| | Do your experiences differ as a result of your breathing at a certain time of the day?<br>Do your experiences differ as a result of your breathing in certain situations?<br>Do your experiences differ as a result of your breathing in certain environments? | -Can you tell me more?<br>How do you react to this/what do you usually do? |
| Health | | |
| | How do you view your health currently?<br>How do you feel your health has been prior to now?<br>Where do you view your health in the future?<br>Do you think your health could improve or worsen? | Why do you view it this way? |
| Patient perspective | | |
| | Is there anything you feel like we haven't covered? | |
| Prompted questions (Only if necessary to elicit adequate responses) | | |

What part of your lifestyle (if any) has changed the most because of your breathing?
I just want to clarify, has your breathing changed any abilities?
Does your breathing pattern vary with any specific activities or factors? Please describe these. (For example, walking, rigorous exercise, eating, or posture?)
Do you feel the change matches the level of effort you exert?
Does your breathing vary with any environmental factors? (for example: cold air, hot air, humidity, smoke, odors).

was initially acceptable but lower than desired at (71%). To improve consensus in the coding system, discrepancies between category systems were discussed and resolved after each sample of the data was co-rated. In discussion, differences were mainly attributed to some overlapping themes in statements. Examples of this included statements that discussed both physical limitations and lifestyle changes such as "I used to enter in small races, and I've stopped doing that because I can't maintain the speed . . . I've gotten a lot slower." In total, raters were provided a sampling of 70% of all the meaningful content from the interviews and ultimately achieved 80% interrater reliability in coding the statements.

## Results

Twenty-seven adult patients diagnosed with ILO provided informed consent to participate in the study (2 males and 25 females) and 26 participants completed an interview (2 males and 24 females). The mean age in the interviewed population was 45 (range 18–72, SD 16.74). The mean age for male participants was 39 (range 37–41, SD 2.83) and for female participants, 45 (range 18–72< SD 17.48). Although there were a disproportionate number of women in the study, the literature describes a predominance of women with diagnosis of ILO [16]. In addition to ILO, just over half (53.8%) of the participants endorsed a diagnosis of asthma, 7.7% were diagnosed with COPD, and 11.5% had obstructive sleep apnea. Fifty percent were

**Table 2. Categorized responses and their frequency across participants.**

| Number of participants who expressed theme N = 26 | Examples of Sub themes | Participant Quotations |
|---|---|---|
| **THEME: Diagnosis and treatment** | | |
| 18 (69.2%) | Presenting to the ER Difficulty diagnosing* Inaccurate Dx or Tx* Understanding or awareness of ILO* | A lot of times it got to the point where I had to go to the ER because I was almost hyperventilating. I don't know, I just feel like with doctors they always have me see other doctors and it kind of goes in circles. I thought it was asthma all these years. . ." I think if a lot of people were aware of other things out there than just asthma or just things like that they could benefit from it. |
| **THEME: Emotional impact** | | |
| 25 (96.2%) | Frustration or emotion * Loss of enjoyment Missing out Just deal/could be worse ILO is scary* Stress d/t ILO Embarrassed or self-conscious* | Very sad, it's huge loss. Sad is such a small word for such a huge loss. It's like a piece of myself is gone. I don't enjoy walking the dog. We live at the top of a hill and I think "okay, going down is great, but coming up. . . Okay I can't breathe. . . .you sit there and watch your friends, they're walking around and they look at you and they think oh should we invite her, shouldn't we invite her. You feel like, like you're not part of it, you know? Most of the time I just try to get through it, I guess. At least I don't feel like I'm drowning, that sort of thing. Because when you have these episodes, it almost seems like it's going to kill you, it's just scary. I try to stay relaxed, but it makes me stressed. [I'm] a little bit more self-conscious . . . I would never go to the gym and get on a treadmill, because I know I'm going to have to do weird, funny breathing. |
| **THEME: Health and prognosis** | | |
| 25 (96.2%) | Negative view of health d/t breathing I'm still healthy* Feeling old or out of shape due to ILO* Other health factors or morbidities* | I feel like I was super super healthy as a child and a teenager, and then it kind of went downhill from there. I think I've always been healthy, and I think I've been proactive to remain healthy. [my health is] good because I'm not struggling. I can go on a run, and I know I can complete a three or four mile run. Even, though, I'm what. . . 42? So, I shouldn't be out here breathing like someone who's 80. They tell me I've got arthritis; I've got fibromyalgia, they tell me this and that I'm just like "really?" I don't want it, take it away. Now there are a lot of other issues going on as well, so this is just compounding it. |
| **THEME: Helpful factors** | | |
| 24 (92.3%) | Air quality helps Laryngeal maneuvers Focus on breathing/control Relaxation Helps* Diagnosis helps Therapy helps* Position/posture help | Cold fresh air [makes it better]. I've never had a trigger with that. I felt like I had to yawn a lot in order to-like, repeatedly, in order to feel like my lungs were properly inflated I swallow a few times and it goes away. I can take a better breath in my nose than I can through my mouth. It seems like I have to be on guard to make sure that I'm okay. I'm more aware of what I'm doing. I just try to breathe and stay calm until the episode ends. And, I feel like, since being diagnosed, I have gotten more of my life back. So, I'm doing all of my activities again. Yeah, I mean, I'm doing everything I used to do, for the most part. [since therapy] I feel much better, I'm not afraid anymore. But as far as posture goes it helps to sit up straight and that's another thing, I need to be more aware of because I'm really good at slouching all day (chuckles). |
| **THEME: Lifestyle** | | |
| 23 (88.5%) | Lifestyle changes* Avoid triggers Avoid or Adjust activity * due to ILO | I want to be able to not opt out of the activities, or to be able to keep my breath and my voice in line when I'm in those stressful situations. I just avoid them [triggers]. If it's going to be muggy outside, I pretty much figure that I'm not going to be spending most of my time outside. But, if it's somebody I don't know very well, I usually just avoid it, and, Tell them an excuse. |
| **THEME: Physical Impact** | | |
| 23/26 (88.5%) | Impact on abilities or physical performance * Dyspnea incongruent with physical fitness struggling more than others Fatigue* Stop/slow down* | I've always been such a capable person, extremely capable, and that. . . It lowers the bar, lowers the bar even further. I'm in shape, I'm a student-athlete, but I have, I've been having a hard time breathing at the moment. When I'm with groups of people like hiking or biking and they're able to go faster than me and I'm behind it bothers me I can't you know keep up like I used to." It just takes everything out of me. Basically because I'm like losing breath. " I get more a short of breath and sometimes I have to stop and rest |

(*Continued*)

**Table 2.** (Continued)

| Number of participants who expressed theme N = 26 | Examples of Sub themes | Participant Quotations |
|---|---|---|
| **THEME: Social impact/Relationships** | | |
| 18 (69.2%) | Concern from others People don't know about ILO Judgment/ lack of understanding Impact on relationships | I do a lot of hiking with my son, and I don't want to put him in a situation where he has to call 911 for his mom. It's very difficult to explain to somebody who doesn't have any experience with asthma, or anyone who has had sort of trouble with their breathing, what it's like you know. Well, I mean, I feel like, just like I'm not respected, I guess, in a way? Like, by my teammates, because, I feel like, and also myself, I can't get this done, so I have to keep doing more It makes you feel bad, and again self-conscious, like . . . I'm not doing things to make my husband happy and spend time with him. |
| **THEME: Descriptions of symptoms and triggers** | | |
| 26 (100%) | Can't breathe in Noisy breathing Symptoms confused with asthma No Symptoms when asleep Cough as a symptom Quick onset and resolution of symptoms Anticipating onset of symptoms Voice/Communication problems* Activity as a trigger* Specific Activity as a trigger Reflux as a trigger Sinus symptoms as a trigger Stress as a trigger Environmental triggers* Triggered eating Triggered talking Time of day/triggers Throat discomfort Throat and/or chest discomfort Dizziness/syncope | I can't breathe in, and get a solid, nice breath into my lungs. It sounds just very wheezy, I guess. Just very, my inhales are very. . . (I guess it just sounds like I'm gasping for air. I think they pretty much think I'm having an asthma attack. I noticed, probably after about a year of these symptoms, that going to sleep, would, like I never had these symptoms when I was sleeping. Like, I would wake up in the morning like, woah, I slept all night and I was fine! Coughing a lot, I would get that after pretty heavy exercise and I would linger with a cough for probably about a half hour. Some days it goes bad I would feel pain in my lungs. Then once I catch my breath I'm fine and then it starts all over again with the next rep [of a workout]. I can kind of feel it coming on before it fully affects me. If I'm in a restaurant, or a place where there are a lot of people, my husband can't hear me, people can't hear me. Because, I can't project my voice. When my breathing issues are triggered, it's anything with an increased rate, walking briskfully [sic], carrying a few things, even walking down stairs, or walking up the stairs. Anytime there is a little bit of an increased heart rate, I feel like my breathing is labored." Heavy cardio is what does it. I have reflux, and when that gets really bad I can definitely feel it getting harder for me to take a breath in. I do know that when I have more sinus symptoms, I do have to concentrate on my breathing a little bit, and I notice that my voice is more gravely-er. But, as far as situations, I don't think that there is a correlation, I can't think of what that would be I'm really aware that it's like the chicken or the egg that I get anxious and stressed, and I can feel myself tighten up." It was worse [in the fall]. I would be short of breath all day long. I get really bad shortness of breath with a lot of environmental air temperatures. Air pollution, cigarette smoke, perfumes, spray cleaners, things like that. Cold weather seems to trigger it easier. It would burn a little bit extra. When your throat is dry it just doesn't feel as good, and it will end up, I find that I have issues with it more often in like drier spaces, spaces that are dusty. Humidity, definitely. I've always felt, I've been around saunas, or even in a pool, I feel like I'm suffocating. I'm more careful about how I chew things. [it can be triggered} if I would talk for a long period of time, and laugh a lot. But, generally, at night is when it's the worst. It feels really like my throat is really high. It feels like my chest and my throat are tightening up. Burning sensation in my chest similar of a bad cold . I start to feel just really tight and more tired overall when it's happening." I get really lightheaded and I kind of feel like I'm going to pass out but I never do. |

* Indicates subtheme shared by >30% of participants.

currently using inhalers. Mental and/or behavioral health diagnoses were noted 42.3% of participants. Included, was report of depression in 30.8% of participants and anxiety in 26.9%.

The interview content was analyzed according to qualitative research methods based on grounded theory [17]. Eight themes in total were identified. Participant's descriptions of the symptoms they experienced and the triggers for their symptoms dominated the interviews. In addition to discussion of symptoms and their triggers (theme 1), seven additional themes were judged to encapsulate the interview material and included: 2) *diagnosis and treatment*, 3) *emotional impact of ILO*, 4) *perception of health and prognosis*, 5) *ameliorating factors*, 6) *influence of ILO on lifestyle*, 7) *the physical impact of ILO*, and 8, *social consequences of ILO*.

## Symptoms and triggers

Twenty-four (92%) of the participants described an environmental trigger for their symptoms. Of the participants who noted environmental triggers, 11 mentioned odors, fumes, and allergens (46%); nine mentioned humidity (38%); and six mentioned cold air (25%). Seventy seven percent of the participants reported symptoms with activity with 8% of those respondents endorsing only specific and often more intense activity as a trigger. Interestingly, discomfort was described explicitly as a symptom by more participants (73%) than dyspnea was (54%). Of those 19 participants who described discomfort as a symptom, discomfort was reported in the throat (47%) and upper chest (26%), with 2 participants describing discomfort in both the throat and upper chest (11%). Five participants described general discomfort as a symptom (26%), and voice or communicative issues were described by more than half of participants.

## Diagnosis and treatment

Most participants described difficulty getting diagnosed and/or treated accurately (65%). These reports often segued into expression of frustration or further emotional impact. Notably, some participants described particular emotional impact from being misdiagnosed when anxiety was the proposed etiology. One participant stated, "I feel like one of the doctors, said, 'Well, this is probably just anxiety.' It's not just anxiety, I know that. To be marked as someone in the ER who is coming in with anxiety, doesn't feel very good. It makes me feel really, really helpless."

## Emotion and ILO

The emotional impact of ILO was a particularly dense topic. All but one participant directly described a negative emotional response to ILO. Sixty-nine percent of participants described frustration or sadness, 50% reported being fearful due to their experiences with ILO. For example, one stated, "when you have these episodes, it almost seems like it's going to kill you, it's just scary." Forty two percent identified embarrassment which was often associated with change in physical performance.

## Health and prognosis

Despite the frequent report of emotional impact, participants often endorsed that they recognized things "could be worse" in the broader landscape of health and wellness, and over half of the participants described "dealing" or "coping" with ILO. Many participants expressed a positive overall view of health and/or a hopefulness regarding their future prognosis while others indicated that they felt old or out of shape due to their dyspnea.

### Lifestyle

Nearly 40% of participants described ways in which avoiding triggers for their dyspnea influenced their lifestyle, noting they avoided certain activities or environments. Examples of this included: "I'm really careful about air quality and whether or not I'm out," and "I avoid places where I know there will be a lot of smokers. I don't go down the laundry detergent aisle in the grocery store."

### Physical impact

Most participants endorsed a physical consequence of ILO (88%). Eighty-five percent reported a change in physical performance marked by an overt change in ability ("I can't you know, keep up like I used to."), or a need to slow down or pause during an activity ("I have to stop and rest."). Nearly 40% of participants described physical fatigue ("It just takes everything out of me"). When participants did describe a change in physical ability, this appeared to be a meaningful aspect of their experience as the theme often was repeated (on average 4 times) throughout an interview.

### Social impact

Social impact due to ILO was mentioned by 69% of participants. Descriptions of social consequences were far ranging in the sample and included avoidance of physical activity with others, and perceived judgement from others (42%). Participants' concerns about judgement alluded to the stigma of physical limitations, noisiness of their breathing, and the way these symptoms could mirror a contagious ailment (coughing, voice change, etc.). In multiple interviews, participants described resentment in relationships due to ILO. In one example of this, a participant stated, "At first, he kind of got panicky, but then we realized that I could get it under control, then it's "Oh here we go. . .Here we go, something else." One very emotional account from a participant described a similar dynamic impacting her relationship "It makes you feel bad, and again self-conscious, like . . . I'm not doing things to make my husband happy and spend time with him."

### Ameliorating factors

A wide range of factors that helped participants manage their dyspnea were described. Although no interview questions prompted this topic specifically, 92% of the participants discussed it and the resulting statements comprised approximately 10% of the meaningful content analyzed in the interviews. 19% of participants indicated that simply being diagnosed with ILO itself was helpful. Report of relief or even a sense of validation in having a diagnosis was a common sub-theme in this leading some participants to cope better with persistent symptoms or return to activity with reduced fear. Some participants mentioned specific techniques that may be employed in therapy like nasal breathing, attention to breathing, balanced posture, physical relaxation and laryngeal maneuvers like swallowing/drinking and yawning, while speech therapy was identified explicitly as a helpful factor by 54% of participants. Therapy was one of the two most frequently cited helpful factors alongside attempting to relax (57%).

## Discussion

The aim of this study was to qualitatively describe patient experiences and perspectives related to ILO so that future treatments and outcome measures can be more specific to the needs of this patient population. Patients who present clinically with ILO report a range of symptoms that can vary based on pattern of triggers and what responses are most helpful [18]. They still

generally share a well-established constellation of symptoms that include dyspnea, change in voice, and discomfort (often localized to the throat and/or upper chest) that are triggered by activity or environmental stimuli [19,20]. The qualitative method of the interviews in this study was selected to promote discussion of patient experience rather than focus on the aspects of ILO that are most salient to clinical care providers. In this setting, a rather robust discussion of the emotional impact of the disorder that had not been widely reported before occurred. This is similar to themes that have emerged recently in the literature on patient perspectives surrounding chronic cough, where the emotional toll of the disorder appeared to be a meaningful aspect of patient experience [21]. For the present investigation, a theme underscoring the importance of appropriate diagnosis and treatment was also common. Although it is well known that patients with ILO often require multiple visits to healthcare professionals [2] and incur meaningful medical costs [2,8] before being accurately diagnosed with ILO, the emotional toll of this as described by participants in this study has not been emphasized in the literature. Further research is necessary to examine these themes specifically and determine how they could impact a patient's self-efficacy and treatment readiness or outcomes.

As clinical care providers, it is important to remain cognizant not only of the physical symptoms of ILO and the strategies used to eliminate them, but of patient experience as a whole. This includes the frustration, stigma, and burnout many patients with ILO experience and the potential for these factors to shape a patient's perceived severity and outcomes. In both the broader medical/surgical literature, and literature specific to voice and ILO, there is evidence to suggest that common objective measures and clinician impression of treatment success and QOL may fail to account for important aspects of the patient perception of outcomes [22–24]. This work underscores the need for clinical care providers to engage in dialogue with patients that addresses significant elements of their perception of the disorder and how it impacts their lives. For example, the validation many patients associate with accurate diagnosis itself draws attention to an important aspect of their experience with the disorder and may highlight an important topic of discussion and education in clinical care visits.

The recruitment of participants from a single tertiary care center is a limitation of the study. Patients seen in these settings have typically had extensive prior medical work-up and may also represent a specific cultural and socioeconomic group. Thus, for many reasons, our participants may have shared experiences that do not reflect a larger population.

It is also possible that allowing participants' treating clinician to serve as the first contact for discussing recruitment may represent a source of bias. Provider-patient relationships are complex in a way that may influence a participant's willingness to consent. In a busy clinical setting, without a dedicated team of assistants and research personnel consistently present on site, deferring all recruitment inquiries to other staff would have likely disrupted clinic flow and been infeasible. While power imbalances between patient and care provider must be considered, the consent process is intended to communicate as compellingly as possible that participation is voluntary and declining to participate does not influence care choices. Additionally, excluding the treating clinician from the interview itself was intended to minimize bias that may be influenced by the patient-client relationship.

Although the qualitative nature of this study is typically considered a limitation, qualitative methods are preferable when attempting to identify specific experiences. Themes captured in this study may provide directions and questions for future work with more quantitative methods.

## Conclusions

Patient perceptions of ILO have not been adequately described in the literature. The present study demonstrates that patients place particular emphasis on the emotional and psychosocial

consequences of the disorder. As such, future efforts to treat ILO and to collect outcomes measures may be inaccurate if they do not address these experiences specifically and proportionally.

## Author Contributions

**Conceptualization:** Katherine M. McConville.

**Formal analysis:** Katherine M. McConville.

**Funding acquisition:** Susan L. Thibeault.

**Investigation:** Katherine M. McConville.

**Methodology:** Katherine M. McConville, Susan L. Thibeault.

**Supervision:** Susan L. Thibeault.

**Writing – original draft:** Katherine M. McConville.

**Writing – review & editing:** Katherine M. McConville, Susan L. Thibeault.

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
