## [Decision Letter · Decision Letter 0]

5 Oct 2023

PONE-D-23-23833Patient perceptions of the impact of Inducible Laryngeal Obstruction on quality of lifePLOS ONE

Dear Dr. McConville,

Thank you for submitting your manuscript to PLOS ONE. After careful consideration, we feel that it has merit but does not fully meet PLOS ONE’s publication criteria as it currently stands. Therefore, we invite you to submit a revised version of the manuscript that addresses the points raised during the review process.

ACADEMIC EDITOR: 

#### Required for Acceptance

1. Provide more descriptive details on the study population, including Exercise-ILO, asthma, COPD, mental health disorders, medications, etc.

2. Re-phrase lines 90-92 to clarify the term 'exclusion diagnosis'.

3. Describe the medical workup for patients, including consultations with ENT specialists and pulmonologists.

4. Clarify the sentence in lines 38-39 of the abstract.

5. State the themes and subthemes directly in the abstract.

6. Revise line 74 in the introduction to better reflect the subject matter.

7. Start the methods section with a statement of study design and setting.

8. Define eligible participants before discussing selection methods.

9. Specify where potential eligible participants were found.

10. Define exclusion criteria for the study.

11. Clarify the transcription process in the methods section.

12. Replace "primary investigator" with initials in the methods section.

13. Clarify the term "7 additional themes" in the results section.

14. Replace the term "subjects" with "participants" or "respondents" in the results section.

15. Include quotes from respondents to validate themes in the results section.

#### Recommended

1. Describe how patients presented, such as MTD/voice, breathlessness, ‘asthma’, chronic cough, irritant-induced, etc.

2. Consider using VCD/ILO or at least mentioning it for clarity.

3. Indicate if any questionnaires were administered to shed light on the (poly)symptomatic burden.

4. Provide frequency or proportion of the various subthemes in the thematic analysis.

5. Create a figure to summarize the work.

6. Clarify the phrase "Participation perception of health and overall prognosis" in the abstract.

7. Revisit the conclusion in the abstract to align with the results.

8. Revise the subheading "semi-structured interviews" to "Data Collection."

9. Describe the role of a "speech-language pathologist" in the study.

10. Clarify the process of devising categories in the methods section.

11. Move lines 124-126 to the methods section under data analysis.

We look forward to receiving your revised manuscript.

Kind regards,

Marcus Tolentino Silva, Ph.D.

Academic Editor

PLOS ONE

 [This work was funded by the Diane Bless chair.  Co-Author Susan Thibeault (ST) is the current chair of this endowment: https://www.surgery.wisc.edu/give/named-chairs-and-professorships/.  ST assisted in manuscript preparation].  

Reviewers' comments:

Reviewer's Responses to Questions

**Comments to the Author**

1. Is the manuscript technically sound, and do the data support the conclusions?

Reviewer #1: Yes

Reviewer #2: No

2. Has the statistical analysis been performed appropriately and rigorously? 

Reviewer #1: N/A

Reviewer #2: N/A

3. Have the authors made all data underlying the findings in their manuscript fully available?

Reviewer #1: No

Reviewer #2: No

4. Is the manuscript presented in an intelligible fashion and written in standard English?

Reviewer #1: Yes

Reviewer #2: Yes

5. Review Comments to the Author

Reviewer #1: Thanks for asking me to review this interesting study. McConville and Thibeault describe qualitative work on a group of 26 patients with ILO and found several themes. The disease burden is striking and multidimensional and characterizing it is highly worthwhile.

A few thoughts:

1. Study population. Can the authors provide more descriptive detail on their study population? Exercise-ILO, asthma, COPD, mental health disorders, medications etc. How did they present? MTD/voice, breathlessness, ‘asthma’, chronic cough, irritant-induced etc. Phenotypes are going to be important to understand mechanisms and tailoring treatments. Giving the reader a sense of the type of patients, and thus their phenotypes, would be quite useful for interpretation and contextualization. For example, at a high level I would anticipate some findings (e.g. ED presentation) would not make sense to the reader who sees an EILO population.

2. Terminology. This paper describes important findings and because patients had vocal fold narrowing, this is really describing VCD/PVFM, and VCD would probably be the term most commonly used/searched for in clinical practice outside of Europe. Because there is such confusion, authors could consider using VCD/ILO or at least mentioning it, the way that has been done in other papers (e.g. 10.1016/j.jaci.2023.06.007).

3a. Diagnosis. On this the ‘gold standard’ see the recent Delphi cited above, it’s well aligned with what authors have described but may be worth considering. Lines 90-92 – from the medical perspective it reads a bit strange saying this an ‘elimination’ or ‘exclusion’ diagnosis but can co-occur with asthma/COPD. When a diagnosis is an ‘exclusion’ diagnosis, all other diagnoses need to be disproven, so by definition it cannot co-occur. Could this be re-phrased.

3b. Did patients in this study really have VCD? Some description of the medical workup e.g. seen & treated by ENT, pulmonologists etc would be useful (some features described do sound like poorly controlled asthma so it’s useful to reassure the reader).

4. Were any questionnaires administered? Can these shed light on the (poly)symptomatic burden?

5. Thematic analysis (table 2) – is it possible to obtain a frequency/proportion of the various subthemes? If 18/18 presented to the ER I’d interpret the data differently to 1/18.

Finally – the themes are nicely explained in the Text and Tables. Can the authors make a Figure to summarize the work?

Reviewer #2: Dear Authors,

Your study seeks to explore an important issue that potential informs improvement of clinical care for the patients with ILO.

The manuscript needs to be improved significantly in order for the work to meet the expected purpose.

1. Abstract

a) In lines 38 - 39; what does this sentence really mean and how is it a result? Is this the first overarching theme; if so, please say it there.

b) It would be easy on the reader if you straight on state the themes and subthemes (if any) that emerged from the analysis process. The semantics here can go into the main text.

c) "Participation perception of health and overall prognosis". What is the message herein? I probably represent the few but important minority who do not understand what this phrase means. Please be more direct and keep it simple.

d) Conclusion: Why do you say the patients seem more impacted by emotional and psychological consequences of ILO? In the results, you seem to state that emotional impact was just one way ILO impacted respondents - why then and how have you reached the comparative stage now?

2. Introduction

a) Good introduction of the subject matter.

b) In line 74, you seem to have missed a word or two: ". . . most consistent, distressful and concerning to the patients with this disorder". Meaningful does not seem to bring out the aspects of importance to clinical care of this patients. And yet, you seem to want to make a contribution to improving the care for and care experience of the patients with ILO!

Please revise accordingly. What I added there is just an example, and a recommendation on the direction required - not the required revision which is at your discretion.

3. Methods

a) Please start this section with statement of the study design and setting. You can then tell us about the ethics later in its own separate section - which I hope is there and detailed enough.

Participants

b) Please first tell the reader who the eligible participants were before delving into how you selected those that participated.

c) Where were the potential eligible participants to be found/accessed? In the hospital; in their homes; if in the hospital, at what stage of their care seeking or which section of the hospital - e.g. clinic, operating theatre etc.

d) So who were excluded from the study? Note that appropriate definition of participants and the inclusion/exclusion criteria for the current study are important in informing readers who may want to transfer your findings; so they make appropriate judgments.

e) The subheading "semi-structured interviews" perhaps refers to "Data collection". Please revise accordingly.

f) Briefly describe a "speech-language pathologist" so that their unique roles and relevance in the data collection process for this study is clear to the readers.

g) Line 98: ". . . and transcribed". Was the transcription done there and then or after sometimes from the days of data collection? And were the transcriptions doe by the same people who collected the data or some other people? Please be explicit on these as they important for informing the readers on the process undertaken and on the elements of rigor and trustworthiness.

h) Line 100: Who is the primary investigator? Please replace this expression "primary investigator" by the initials of that person you are making reference to.

i) Line 100: Did he/she devise categories right away as he/she read the transcripts - or were there some other processes undertaken before devising the categories? Please be clear on what actually happened and on the how. Open up that black box please.

j) Lines 98 - 115 are describing the data analysis process. However, I am afraid that I have failed to follow and understand the process being described. Please make the description of the data analysis process simple and accessible to ordinary mortals.

Results

a) Lines 124 - 126 belong to the methods section; specifically under data analysis. And you bring in here issue of Grounded Theory while elsewhere you report following the Burnard's 14 step method. Are the two approaches synonymous? If yes, then say so and why you are referring to them interchangeably.

b) Line 126: What do you mean 7 additional themes? Where are the other earlier theme and subthemes (if any) before you start talking about the additional themes? We wish to see written the themes and subthemes - the earlier and the 7 additional ones. Why are they additional anyway? Why are they not part and parcel of the rest of the themes? What is different about them or why did they emerge later so to attract the expression " additional"?

c) Lines 131 - 132: are they subjects or participants/respondents? The 2002 CIOMS report on research involving human subjects as participants recommends use of the term participants for quantitative and respondents for qualitative studies. You may wish to replace the rather derogatory term subject with respondents.

d) I stopped review at line 197. I have seen two themes and none of them have adequately included under them respondents' voices as quotes that serve to validate the themes. I also do not see the 7 additional themes explained with their respect quotes. The quote form the backbone for the results and they add credence to the study - validating the themes but also contribute to the trustworthiness, rigor and credibility of the study.

I will be glad to review the revised manuscript that I expect to have the quotes and code of respondents attached to the quotes.

Kind regards

6. PLOS authors have the option to publish the peer review history of their article (what does this mean?). If published, this will include your full peer review and any attached files.

Reviewer #1: No

Reviewer #2: **Yes: **Amos Deogratius Mwaka

---

## [Author Response · Author response to Decision Letter 0]

19 Jan 2024

We have enclosed a letter with a complete list of responses/revisions made for each comment received. We again thank you for your time and expertise in your review and your consideration of this work for publication.

---

## [Decision Letter · Decision Letter 1]

29 Feb 2024

PONE-D-23-23833R1Patient perceptions of the impact of Inducible Laryngeal Obstruction on quality of lifePLOS ONE

Dear Dr. McConville,

Thank you for submitting your manuscript to PLOS ONE. After careful consideration, we feel that it has merit but does not fully meet PLOS ONE’s publication criteria as it currently stands. Therefore, we invite you to submit a revised version of the manuscript that addresses the points raised during the review process.

**ACADEMIC EDITOR: **

** Required Changes

* Manuscript Format and Structure

- Revise the manuscript to follow the journal's specific formatting and structural guidelines.

* Methods Section

- Clearly delineate and elaborate on the study setting, design, participant recruitment, data collection, and data analysis methods.

* Abstract Revisions

- Use specific, scholarly language in the abstract to clearly convey study methods, results, and significance.

- Specifically describe the content analysis process, including the role of raters and consensus formation.

* Results Section

- Clearly present and contextualize themes and subthemes derived from the data.

** Recommended Changes

- Ensure submission of a single, coherent revised version of the manuscript that incorporates all changes requested by reviewers.

- Modify comparative analysis statements in the Results section of the abstract to focus on direct presentation of findings.

- Use language that is accessible and appropriate for an academic audience, ensuring that the study's findings are easily understood.

- Strengthen the connection between the study's conclusions and the results presented, particularly regarding the social consequences of ILO diagnosis.

- Consider professional editing to ensure the manuscript meets the highest standards of academic writing and clarity.

We look forward to receiving your revised manuscript.

Kind regards,

Marcus Tolentino Silva, Ph.D.

Academic Editor

PLOS ONE

Journal Requirements:

Reviewers' comments:

Reviewer's Responses to Questions

**Comments to the Author**

1. If the authors have adequately addressed your comments raised in a previous round of review and you feel that this manuscript is now acceptable for publication, you may indicate that here to bypass the “Comments to the Author” section, enter your conflict of interest statement in the “Confidential to Editor” section, and submit your "Accept" recommendation.

Reviewer #1: All comments have been addressed

Reviewer #2: All comments have been addressed

2. Is the manuscript technically sound, and do the data support the conclusions?

Reviewer #1: Yes

Reviewer #2: Partly

3. Has the statistical analysis been performed appropriately and rigorously? 

Reviewer #1: Yes

Reviewer #2: N/A

4. Have the authors made all data underlying the findings in their manuscript fully available?

Reviewer #1: No

Reviewer #2: Yes

5. Is the manuscript presented in an intelligible fashion and written in standard English?

Reviewer #1: Yes

Reviewer #2: No

6. Review Comments to the Author

Reviewer #1: Comments addressed, thank you. No further comments. Although the authors are right that "ILO" is supplanting "VCD", many North Americans (including recent peer reviewed work), primary care docs, many patients, and most critically, people who are stumbling across the field for the first time, will look up or use "VCD". I still strongly suggest putting "VCD" in the abstract to enhance the visibility of the work and then you can clarify ILO at first mention. Otherwise a large portion of potential readership may miss your very important findings!

Reviewer #2: Dear Authors,

Again, thank you for improving on this manuscript. You probably want to follow the scientific journal format. For example, under methods, the section opens with a statement on ethics approval rather than study setting and design. I also find certain important aspects of methods either missing or clumped where they should not be; the section labelled “semi-structured interviews” has in it (1) participants’ recruitment producers, (2) data collection, (3) data analysis. This makes the review process challenging because matters are where they should not be and they are brief, leaving readers unable to see through what exactly was done and how they were done. I find two clan copies and one copy with track changes. The second clean copy appears to reflect the revised manuscript. But it still not following the manuscript style. For example, data collection subheading has in it data analysis as well.

I therefore stopped my review at the methods section. Otherwise, find my specific comments on the abstract below.

Abstracts – background

You may want to take note of the expression, “…in a meaningful proportion…” It is difficult to understand what you exactly mean. Kindly use formal scholar expression to say the same.

Abstract – Methods

The statement, “Transcription underwent content analyses across multiple raters until consensus regarding analyses was reached”. It is again quite challenging for me to understand the message you are communicating here. You may want to revise it. So did you use content analysis approach to the data/transcript? Was the consensus on codes or themes? When you talk of raters, it is challenging to get what you mean in the sense of qualitative analyses approaches.

Abstract – Results

The statement, “Consistent with literature . . . with ILO…” This is the style for discussion where you make comparisons of your findings with existing literature and yet here you are reporting results of this study. You may want to revise accordingly.

Lines 38 – 39; “Patients description . . . in all interviews”. What is the message in this statement? Kindly state it differently so that it becomes easy for an ordinary reader like myself to peak up the message. You may for example want to provide exactly what their descriptions of symptoms were i.e. you provide the themes and or subthemes on symptoms.

Again, lines 40 – 41; “Seven additional themes . . . the interview material”. The immediate question is – additional to which ones? And one does not see fore to this statement any descriptions of themes! Kindly present the themes and subthemes under your results section, and among the themes/subthemes there will be those on symptoms at presentation.

Line 42; “. . . participation perception . . .” This expression is not regularly used in ordinary academic discourse. Kindly rephrase so that you clearly bring out what you indeed wish to communicate. In the current form, an ordinary reader like me fails to get the message.

Conclusions: Your emphasis is on the consequences of ILO diagnosis, especially social consequences. That is alright. However, you may want to have issues on consequences directly or indirectly alluded to in the result section, and should appear to carry a high significance so that it becomes the major issue in the conclusion.

Kind regards

7. PLOS authors have the option to publish the peer review history of their article (what does this mean?). If published, this will include your full peer review and any attached files.

Reviewer #1: No

Reviewer #2: No

---

## [Author Response · Author response to Decision Letter 1]

29 Mar 2024

Dear Esteemed Reviewers,

We thank you for your continued time and attention in the review of this work. Your comments and insight have guided us in the revision process such that we hope the result is a manuscript that reflects your feedback and would better communicate our work, findings and their relevance to this journal’s readership.

The remaining reviewer comments and our responses are included below and are also attached in a separate document.

With Gratitude,

Katherine McConville & Susan L. Thibeault

Reviewer #1

• Comments addressed, thank you. No further comments. Although the authors are right that "ILO" is supplanting "VCD", many North Americans (including recent peer reviewed work), primary care docs, many patients, and most critically, people who are stumbling across the field for the first time, will look up or use "VCD". I still strongly suggest putting "VCD" in the abstract to enhance the visibility of the work and then you can clarify ILO at first mention. Otherwise, a large portion of potential readership may miss your very important findings!

Response: We truly appreciate your review. Thank you for your thoughtful comments and guidance!

Reviewer #2:

• You probably want to follow the scientific journal format. For example, under methods, the section opens with a statement on ethics approval rather than study setting and design.

Response: Thank you for your feedback. After looking over several recent articles in this publication, we have changed the placement of the ethics approval statement to mirror its placement in similar published articles. We have also revised our user of headers to better follow the specified format.

• I also find certain important aspects of methods either missing or clumped where they should not be; the section labelled “semi-structured interviews” has in it (1) participants’ recruitment producers, (2) data collection, (3) data analysis. This makes the review process challenging because matters are where they should not be and they are brief, leaving readers unable to see through what exactly was done and how they were done. I find two clan copies and one copy with track changes. The second clean copy appears to reflect the revised manuscript. But it still not following the manuscript style. For example, data collection subheading has in it data analysis as well. I therefore stopped my review at the methods section. Otherwise, find my specific comments on the abstract below.

Response: Thank you for pointing this out. We have reorganized some of the content in the “Materials and methods” section and revised our user of headers and the format of these headers to follow the specified format and hopefully improve readability.

• Abstracts – background

You may want to take note of the expression, “…in a meaningful proportion…” It is difficult to understand what you exactly mean. Kindly use formal scholar expression to say the same.

Response: Thank you for the thoughtful feedback. We have changed the description to “noteworthy” and hope this is acceptable. Our rationale for the use of “meaningful” had been that we felt it conveyed that this specific disorder contributes to dyspnea in a noteworthy proportion of patients with dyspnea. “Significant” was avoided as we are not speaking to statistical significance and wish to prevent confusion. “Large” was not selected as it might infer a majority and the literature shows that this does not make up a majority of these cases. For example, 22% of patients with frequent emergency room visits due to sudden onset dyspnea are suspected to have ILO. 

• Abstract – Methods: The statement, “Transcription underwent content analyses across multiple raters until consensus regarding analyses was reached”. It is again quite challenging for me to understand the message you are communicating here. You may want to revise it. So did you use content analysis approach to the data/transcript? Was the consensus on codes or themes? When you talk of raters, it is challenging to get what you mean in the sense of qualitative analyses approaches.

Response: We value your feedback and expertise. We have included more detail in the abstract that identifies the published method we used. We hope this provides more clarity in the abstract. The methods are described in more detail in the body of the manuscript. 

• Abstract – Results: The statement, “Consistent with literature . . . with ILO…” This is the style for discussion where you make comparisons of your findings with existing literature and yet here you are reporting results of this study. You may want to revise accordingly.

Response: Thank you for the feedback. The preponderance of female cases of ILO is so notorious in the literature on this disorder that it seemed strange not to acknowledge that this was expected. Your feedback makes clear that it is not appropriate in the abstract, and it has been removed.

• Lines 38 – 39; “Patients description . . . in all interviews”. What is the message in this statement? Kindly state it differently so that it becomes easy for an ordinary reader like myself to peak up the message. You may for example want to provide exactly what their descriptions of symptoms were i.e. you provide the themes and or subthemes on symptoms.

Response: We have clarified that “symptoms and triggers” was judged to be a theme in the interview data and why. We appreciate this feedback and hope it improves the readability of this section. 

• Again, lines 40 – 41; “Seven additional themes . . . the interview material”. The immediate question is – additional to which ones? And one does not see fore to this statement any descriptions of themes! Kindly present the themes and subthemes under your results section, and among the themes/subthemes there will be those on symptoms at presentation.

Response: Thank you for the feedback. We believe the revisions made in lines 38-39 now help clarify this as “symptoms and triggers” was labeled the first theme. 

• Line 42; “. . . participation perception . . .” This expression is not regularly used in ordinary academic discourse. Kindly rephrase so that you clearly bring out what you indeed wish to communicate. In the current form, an ordinary reader like me fails to get the message.

Response: We have clarified that “health and prognosis” was a theme extracted from the interviews. The expression triggering confusion was intended to convey that the participant’s perspectives regarding health and prognosis were a theme. We hope the new wording better conveys this and we thank you for the feedback. 

• Conclusions: Your emphasis is on the consequences of ILO diagnosis, especially social consequences. That is alright. However, you may want to have issues on consequences directly or indirectly alluded to in the result section, and should appear to carry a high significance so that it becomes the major issue in the conclusion.

Response: Thank you for the feedback. We hope that some revisions made to both the results and the conclusions sections address this concern.

---

## [Decision Letter · Decision Letter 2]

25 Apr 2024

PONE-D-23-23833R2Patient perceptions of the impact of Inducible Laryngeal Obstruction on quality of lifePLOS ONE

Dear Dr. McConville,

Thank you for submitting your manuscript to PLOS ONE. After careful consideration, we feel that it has merit but does not fully meet PLOS ONE’s publication criteria as it currently stands. Therefore, we invite you to submit a revised version of the manuscript that addresses the points raised during the review process.

**ACADEMIC EDITOR: **Clearly state the study design (e.g., phenomenology, case study, etc.).Move the study design information to the beginning of the materials and methods section, before data collection details.Address why recruitment was conducted by treating clinicians instead of the research team.Discuss potential power imbalances between patients and clinicians regarding participation and consent.Analyze how these power imbalances might influence the quality of the data.Ensure the ethical considerations, currently placed under "semi-structured interviews," are clearly addressed and appropriately positioned within the manuscript.==============================

We look forward to receiving your revised manuscript.

Kind regards,

Marcus Tolentino Silva, Ph.D.

Academic Editor

PLOS ONE

Journal Requirements:

Reviewers' comments:

Reviewer's Responses to Questions

**Comments to the Author**

1. If the authors have adequately addressed your comments raised in a previous round of review and you feel that this manuscript is now acceptable for publication, you may indicate that here to bypass the “Comments to the Author” section, enter your conflict of interest statement in the “Confidential to Editor” section, and submit your "Accept" recommendation.

Reviewer #1: All comments have been addressed

Reviewer #2: All comments have been addressed

2. Is the manuscript technically sound, and do the data support the conclusions?

Reviewer #1: Yes

Reviewer #2: Yes

3. Has the statistical analysis been performed appropriately and rigorously? 

Reviewer #1: N/A

Reviewer #2: N/A

4. Have the authors made all data underlying the findings in their manuscript fully available?

Reviewer #1: Yes

Reviewer #2: Yes

5. Is the manuscript presented in an intelligible fashion and written in standard English?

Reviewer #1: Yes

Reviewer #2: Yes

6. Review Comments to the Author

Reviewer #1: Nil further comments. Thank you. Revision is satisfactory and I think it is suitable for publication.

Reviewer #2: Dear Authors,

This manuscript has been greatly improved. Congratulations. You can improve on it further by revising it alongside the journal format and style especially the materials and methods section. The ethical considerations is included under "semi-structured interviews". The study design is missing. What is this study design? Is it phenomenology? The first statement under materials and methods is on data collection when it should indeed start with study design and setting.

Then, under participant recruitment and selection, you state that recruitment was conducted by the treating clinician. Why was recruitment done by clinicians and not the research team? You may want to discuss the power imbalance between patient and treating physicians with respect to accepting participation and consent as well as discuss the influence of this power imbalances on the quality of the data.

I am quite comfortable with the results and discussions sections .

I wish you the very best.

Regards

7. PLOS authors have the option to publish the peer review history of their article (what does this mean?). If published, this will include your full peer review and any attached files.

Reviewer #1: No

Reviewer #2: **Yes: **Amos Deogratius Mwaka

---

## [Author Response · Author response to Decision Letter 2]

25 Jun 2024

Our reviewer responses are also attached as a document:

Reviewer #1:

 Nil further comments. Thank you. Revision is satisfactory and I think it is suitable for publication.

Response: Thank you again for your time and attention to this work.

Reviewer #2: 

This manuscript has been greatly improved. Congratulations. You can improve on it further by revising it alongside the journal format and style especially the materials and methods section. 

Response: Thank you, please see additional responses and revisions in an effort to achieve this. 

The ethical considerations is included under "semi-structured interviews". 

Response: Thank you for pointing this out. Ethical considerations are now included under a unique heading. We believe this new section also adds some context that may address your concerns about recruitment (though this is further addressed in the discussion section now based on your feedback).

The study design is missing. What is this study design? Is it phenomenology? The first statement under materials and methods is on data collection when it should indeed start with study design and setting.

Response: Thank you for your feedback. We have clarified study design.

Under participant recruitment and selection, you state that recruitment was conducted by the treating clinician. Why was recruitment done by clinicians and not the research team? You may want to discuss the power imbalance between patient and treating physicians with respect to accepting participation and consent as well as discuss the influence of this power imbalances on the quality of the data.

Response: Thank you for raising this excellent point. In response to your feedback, we have clarified that the treating clinician recruited participants but did not conduct the interview out of concern this would bias responses. 

Our discussion section now further explains that recruitment patterns had to work around our busy clinic model with limited resources for a constant onsite research support team. 

 Your additional points about power imbalance are well-taken and we have tried to address these further in the discussion as well.

I am quite comfortable with the results and discussions sections.

Response: We sincerely appreciate your feedback.

---

## [Editor Report · Decision Letter 3]

27 Jun 2024

Patient perceptions of the impact of Inducible Laryngeal Obstruction on quality of life

PONE-D-23-23833R3

Dear Dr. McConville,

We’re pleased to inform you that your manuscript has been judged scientifically suitable for publication and will be formally accepted for publication once it meets all outstanding technical requirements.

Kind regards,

Marcus Tolentino Silva, Ph.D.

Academic Editor

PLOS ONE
---

## [Editor Report · Acceptance letter]

8 Jul 2024

PONE-D-23-23833R3 

PLOS ONE

Dear Dr. McConville, 

I'm pleased to inform you that your manuscript has been deemed suitable for publication in PLOS ONE. Congratulations! Your manuscript is now being handed over to our production team.

Kind regards, 

on behalf of

Prof Marcus Tolentino Silva 

Academic Editor

PLOS ONE